# Effect of *Lactococcus lactis* JNU 534 Supplementation on the Performance, Blood Parameters and Meat Characteristics of *Salmonella enteritidis* Inoculated Broilers

**DOI:** 10.3390/microorganisms13030525

**Published:** 2025-02-27

**Authors:** Listya Purnamasari, Joseph F. dela Cruz, Dae-Yeon Cho, Kwang-Ho Lee, Sung-Min Cho, Seung-Sik Chung, Yong-Jun Choi, Jun-Koo Yi, Seong-Gu Hwang

**Affiliations:** 1School of Animal Life Convergence Science, Hankyong National University, Anseong 17579, Republic of Korea; listyap.faperta@unej.ac.id (L.P.);; 2Department of Animal Husbandry, Faculty of Agriculture, University of Jember, Jember 68121, Indonesia; 3Department of Basic Veterinary Sciences, College of Veterinary Medicine, University of the Philippines Los Baños, Laguna 4031, Philippines; jfdelacruz@up.edu.ph; 4Elimland Co., Ltd., Namyangju 12106, Republic of Korea

**Keywords:** *Lactococcus*, pathogenic bacteria, probiotics, feed additives

## Abstract

Salmonellosis in broilers is a disease with considerable economic implications for the poultry industry. As a foodborne illness, it also poses a public health risk due to potential cross-contamination. Probiotics have been proposed as alternative feed additives aiming to enhance growth, livestock productivity, and overall health. This study investigated the dietary impact of *Lactococcus lactis* JNU 534 on growth performance, blood characteristics, internal organ weight, and meat quality in broilers inoculated with *Salmonella enteritidis* (SE). A total of 96 one-day-old Arbor Acres broiler chickens, comprising both sexes, were challenged with SE and randomly assigned into two treatment groups and housed in eight pens (four pens per each treatment, with 12 birds per pen). They were fed a commercial broiler diet for 35 days. The two dietary treatment groups consisted of a control group receiving commercial feed, and a treatment group receiving commercial feed supplemented with 0.3% *L. lactis* JNU 534. Probiotic supplementation significantly improved average body weight gain, feed efficiency, and carcass yield compared to the control group (*p* < 0.05). Notably, the abdominal fat pad was significantly reduced in the probiotics group (*p* < 0.05). Meat quality assessments revealed no significant differences between the groups in terms of meat pH, cooking loss, drip loss, and water-holding capacity. These findings suggest that *L. lactis* JNU 534 is a promising candidate to mitigate the negative effects of *Salmonella* on growth performance in commercial broiler farms, without adversely affecting health. Extending the research to other types of livestock could help confirm its wider use as an alternative to antibiotics.

## 1. Introduction

The growing demand for food production increases the challenge of controlling microbiological risks within supply chains [1], raising concerns about food safety, particularly regarding *Salmonella* infections [2]. *Salmonella* is a Gram-negative bacterium, extensively distributed throughout the environment and often located in the gastrointestinal system. Salmonellosis is the most common foodborne bacterial disease in both humans and animals, and it ranks as the second most often reported foodborne gastrointestinal illness in people in EU nations [3]. The consumption of contaminated food such as broiler meat and its derivates products is an important route of this infection. In broilers, it is a significant disease that not only causes high economic costs in the poultry industry but also impacts public health through cross-contamination [4]. *Salmonella*, being an enteric pathogen, reaches the intestine via oral ingestion (horizontal transmission) from contaminated environments, feed, and water.

Although there are more than 2500 *Salmonella* serotypes in nature, only 20 of them are responsible for more than 82% of human infections, with the majority linked to *Salmonella Enteritidis*, *Typhimurium*, *Newport*, and *Heidelberg*. One of the common serovars isolated from broilers is *Enteritidis* [5]. Even a very low infective dose of *Salmonella Enteritidis*, as low as 1–5 bacteria cells, can lead to infection in day-old chicks, lead to high chick mortality and decrease egg production in adult chickens [6]. Effective antimicrobial agents, such as antibiotics, are essential as antibiotic growth promotors and infectious disease control methods including to control *Salmonella* infections. However, the inappropriate use of antibiotics can lead to antibiotic residues and resistance in pathogens, which can be harmful to both animals and humans [7].

For several decades, the use of probiotics to control and prevent pathogenic bacteria has demonstrated beneficial effects. Numerous studies have shown that optimizing gut microbiota is a key strategy to alleviate foodborne pathogen infections. Beneficial microbiota in the animal intestine can enhance the structure of the intestinal mucosa and boost the immune response. Additionally, the active compounds they produce help to suppress the growth of harmful pathogens. Probiotics exert their positive effects by improving intestinal barrier functions, modulating the host’s immune system, and competitively excluding pathogens [8]. As feed additives, probiotics can serve as alternatives to antibiotics, optimizing growth, livestock productivity, and health [9,10]. Among the available probiotics, a type of lactic acid bacteria such as *Lactococcus* spp., well-known Gram-positive bacterium is a common probiotic used in the poultry industry [11]. Organic acids, hydrogen peroxide, bacteriocin, and other metabolites produced by lactic acid bacteria, play a key role in their antimicrobial capabilities.

*Lactococcus lactis* is a homofermentative bacterium, spherical-shaped, typically used in the fermentation of products such as silage [12], dairy products/milk [13], and kimchi [14]. *L. lactis* is recognized for its various beneficial effects on hosts, including immune modulation, enhanced digestion, and the reduction in diarrhea in animals [11]. The antimicrobial substances produced by *L. lactis*, such as lactic acid and bacteriocins, can effectively inhibit or kill pathogenic bacteria including *S. Enteritidis*, *E. coli* and *S. aureus*. Recent studies indicate that *L. lactis* reduced methane production in ruminants due to its ability to inhibit rumen methanogens [15]. *Lactococcus lactis* JNU 534 was first isolated from kimchi [14], produces a new bacteriocin, a natural preservative safe for use in food in several countries [16].

Among the antibacterial compounds produced by lactic acid bacteria, bacteriocin is a natural proteinaceous substance that inhibits the growth of pathogenic bacteria within the gastrointestinal tract. Since bacteriocins are polypeptides (small-sized proteins), it is assumed that proteolytic enzymes in the stomach and other parts of the intestinal tract can hydrolyze them, rendering them inert [17]. This bacteriocin is also non-toxic, making it suitable as a food or feed additive [16]. Although many studies have investigated probiotics in poultry, *L. lactis* JNU 534 has not been extensively studied. This study aims to determine the effect of dietary *L. lactis* JNU 534 on growth performance, internal organ weight, and meat characteristics in broilers inoculated with *Salmonella enteritidis* (SE).

## 2. Materials and Methods

This study was conducted at conducted from March to December 2023. The chicken were raised at an animal research farm facility in Anseong, Republic of Korea. All animal care procedures and management were approved by the School of Animal Life Convergence Science, Hankyong National University Animal Care and Use Committee (Approval Number: Hankyong 2023-1). The blood and meat analyses were conducted at the Applied Biochemistry Laboratory, Hankyong National University, Republic of Korea.

A total of 96 Arbor Acres, one-day-old broiler chickens comprising both sexes were challenged with *Salmonella enteritidis* at a concentration of 1 × 10^9^ CFU/mL on the third day orally. They were then randomly assigned to two treatment groups, housed in a total of 8 pens, with 4 pens per treatment and 12 birds per pen. The birds had free access to water and feed. The visual excreta were checked for proving that all chicken were infected. Other symptoms were weakness, loss of appetite, and poor growth.

The experimental diets were administered for 35 days, with the dietary treatments consisting of a control group receiving commercial feed and a treatment group receiving commercial feed supplemented with 0.3% *L. lactis* JNU 534 (1–2 × 10^9^ CFU/g). The feed form was crumble and the *L. lactis* JNU 534 was supplemented as a top dressing to create respective diets. The *L. lactis* JNU 534 was manufactured by Elimland Co., Ltd., Gyeonggi-do, Republic of Korea. This research used 3-phase feeding program that included starter (0–13 d), grower (14–27 d) and finisher (28–35 d). The composition of the basal diet used in the experiment is presented in Table 1.

All birds were weighed weekly using a sensitive balance to measure body weight gain. Feed intake was determined weekly by calculating the difference between the quantity of the given feed and the leftovers. The feed conversion ratio was subsequently calculated based on body weight gain and feed consumption during the experimental period. At the end of the trial, two hens from each replicate (totaling eight birds per treatment) were randomly selected, fasted overnight, and then sacrificed for full blood sampling. These birds were used for carcass analysis. After removing the skin, feathers, viscera, head, and feet, the carcass weight was measured. Carcass cut-up parts, including breast, drumstick and thigh were separated and weighed from the eviscerated carcass. All carcass traits and relative organ weights were expressed as a percentage of the live weights. This method was measured according to Wandita et al. [18].

At the end of the trial, two hens from each replicate (a total of eight birds per treatment) were selected for serum collection. Ethylenediaminetetraacetic acid (EDTA), was used as an anticoagulant during the collection process. Blood samples were drawn from the brachial wing vein into vacutainer tubes and centrifuged for 10 min at 275× *g* at room temperature within two hours of collection. The separated serum was frozen (−20 °C) until the biochemical analysis. The following parameters were determined from the blood serum samples (*n* = 8) by Seegene Medical Foundation (Seoul, 04805, Republic of Korea): bilirubin, cholesterol, triglyceride, free fatty acid, glucose, white blood cell (WBC), red blood cell (RBC), hemoglobin (Hb), hematocrit (Hct), segment, lymphocyte, monocyte and basophil.

The breast, thigh and drumstick meat samples were kept at 4 °C for 24 h. The pH, water holding capacity (WHC), cooking loss, and drip loss were determined to assess and compare the physicochemical characteristics of control and treatment group broiler meat. A total of 10 g of meat was weighed and 90 mL of distilled water was added before chopping the meat with a meat chopper. The pH was measured using a pH meter. The device was calibrated with buffers of pH 4.0 and pH 7.0. The WHC of the samples was measured using centrifugal force. Whatman filter paper (No. 3) was cut into quarters, and a 1.5 g meat sample (W0) was weighed. The meat sample was covered with three layers of the quartered filter paper, placed in a 50 mL tube, and centrifuged at 3000 rpm. After centrifugation, the sample was weighed again (W1). WHC (%) was calculated as the ratio of weight loss of the sample during centrifugation, to that of the original liquid. The WHC was calculated using the following formula: WHC (%) = [100 − {(W0 − W1)/W0} × 100]. Cooking loss (CL) was measured by weighing 20 g of sample (W0), then put in a 50 mL tube, and centrifuging it at 275× *g* for 10 s, followed by heating in a water bath at 65 °C for 30 min. After cooling, the cooked sample was reweighed (W1). The CL was calculating using the following formula: CL (%) = [(W0 − W1)/W0 × 100].

Data were first tested for the normality of the residual using the Shapiro–Wilk test and the homogeneity of variance was verified using Levene’s test considering each pen as the experimental unit. Data are shown as the mean with the result of standard error of means and *p*-value. Differences between groups were analyzed by the independent sample T-test. The level of statistical significances was set at *p* < 0.05. The statistical software package SPSS 24.0 (SPSS, Inc., Chicago, IL, USA) was used for the analysis.

## 3. Results

Table 2 presents the performance of broiler challenged with *Salmonella enteritidis* and supplemented with *Lactococcus lactis* JNU 534 in their diet. The result indicated that the average body weight gain and feed efficiency were significantly improved by probiotics (*p* < 0.05) compared to the control group. However, there was no significant difference (*p* > 0.05) in feed intake between the groups.

The study showed that *L. lactis* JNU 534 did not significantly affect (*p* > 0,05) the biochemical blood parameters (white blood cells, red blood cells, hemoglobin, hematocrit, segmented neutrophils, lymphocytes, monocytes, eosinophils, and basophils), as indicated in Table 3. This finding underscores the relative safety of *L. lactis* JNU 534 for broilers. Table 3 also showed that the cholesterol and free fatty acid levels in the blood of chickens supplemented with *L. lactis* JNU 534 were significantly lower than those in the control group (*p* < 0.05).

Table 4 showed that the carcass yield and breast yield were significantly influenced by probiotics (*p* < 0.05) compared to the control group. Notably, the probiotics group exhibited significantly lower abdominal fat pad and ileum weight than the control group (*p* < 0.05).

As presented in Table 5, the meat characteristics, including meat pH, cooking loss, drip loss, and water-holding capacity, were not significantly affected (*p* > 0.05) by *L. lactis* JNU 534 supplementation in broilers challenged with *Salmonella enteritidis*.

## 4. Discussion

In recent decades, the role of probiotics in enhancing poultry growth performance and health has been well documented. Probiotics are closely linked to the host’s immune system. A critical challenge lies in selecting potential probiotic strains that can successfully function in the gastrointestinal tract (GIT) and withstand various stressors [19]. Probiotics can reduce the colonization rate of pathogenic bacteria on the intestinal wall, improve intestinal morphology, stimulate the immune system, enhance metabolic function, and reduce the risk of infection [20]. The synergistic effects of intestinal microstructure and intestinal microbes influence digestion and absorption, regulating body weight and feed intake [21]. *Lactococcus lactis* improves intestinal morphology by increasing the absorptive surface area in the small intestine, leading to better nutrient availability [22]. *Lactococcus*-based probiotics are also known for their antimicrobial and antioxidant activities, which are particularly beneficial for broilers challenged with pathogenic bacteria such as *Salmonella*, the causative agent of salmonellosis [23,24].

Pathogenic bacteria can impede protein breakdown, thereby reducing dietary protein efficiency. Lactic acid bacteria, as probiotics, competitively exclude pathogenic bacteria, increase villi height, and improve amino acid utilization [25]. Consequently, the beneficial effects of *L. lactis* JNU 534 on broiler health and performance are evident through increased feed absorption efficiency. The basis for the 0.3% addition of *L. lactis* JNU 534 due to the antibiotic growth promotor banned nowadays is in the range of 2.5–3% on feed. We tested this for *L. lactis* JNU 534 by substituting the use of antibiotic to prevent the salmonella infection. *L. lactis* also have antibacterial compounds called bacteriocin [14].

Bacteriocin is a natural proteinaceous substance that inhibits the growth of pathogenic bacteria. Several barriers may be involved and affect their stability and biological activity. Since bacteriocins are polypeptides (small-sized proteins), it is assumed that proteolytic enzymes in the stomach and other parts of the intestinal tract can hydrolyze them, rendering them inert [26]. A new bacteriocin-producing lactic acid bacteria isolated from kimchi was identified as *L. lactis* JNU 534, presenting preservative properties for foods of animal origin [14]. Several studies have shown the protective effects of LAB bacteriocins on the gastrointestinal tract via eliminating pathogens such as *Salmonella enteritidis*, *Listeria monocytogenes*, *Clostridium difficile*, *Staphylococcus aureus* [27] or supporting the gut from bacterial colonization [26].

Regarding the productive performance Table 2, in the finisher phase, the result indicated that the average body weight gain, feed intake, and feed efficiency were significantly improved by *L. lactis* JNU 534. An improvement impact of probiotic in this study on the growth rate may be attributed to enhanced intestinal health and the ability of probiotics to secrete enzymes such as amylase, protease, and lipase, leading to better nutrient digestion and absorption. This result aligned with Zhang et al. [28], who found a significant effect for 1% probiotics (*Lactobacillus casei*, *L. acidophilus* and *Bifidobacterium*) on broiler performance.

There is a notable decrease in serum cholesterol chickens fed a diet supplemented with *L. lactis* JNU 534. According to Ashayerizadeh et al. [29] and Axling et al. [30], probiotics can decrease cholesterol levels by reducing the hepatic mRNA expression of acetyl-CoA carboxylase, the enzyme that catalyzes the rate-limiting step in fatty acid biosynthesis. This reduction represents decreased de novo lipogenesis and reduced body fat. Additionally, lactic acid-producing bacteria may interfere with cholesterol absorption in the gut by deconjugating bile salts or directly assimilating cholesterol [31], as well as reducing or inhibiting the expression levels of Niemann–Pick C1-like 1 (NPC1L1) protein on the surface of enterocytes, thereby decreasing cholesterol absorption [32]. Other research has reported a decrease in the total cholesterol and triglyceride levels in the blood [25], which correlates with reducing cholesterol and fat levels in poultry meat [33].

The glucose level of chicken supplemented with *L. lactis* JNU 534 was significantly higher than the control group (*p* < 0.05). Blood glucose reflects a balance of the amount of glucose absorbed from the small intestine, the glucose released by the liver into the blood, and that going from the blood directly into the body’s cells [34]. Increased glucose concentrations in the blood may enhance the passive transport of glucose from the intestinal lumen to balance glucose levels inside and outside the cells [35].

Animal growth depends on the digestion and absorption of nutrients in the gastrointestinal tract. One benefit of poultry production is the ability to achieve a high slaughter yield in a very short time. The most desired part of the carcass is the breast muscle. in the present study, the breast muscle of a carcass with *L. lactis* JNU 534 is higher, accounting for about 19.45% of live weight. Fast growth and muscle development are essential from an economic perspective. The carcass yield of broilers exceeded 75%, indicating superior meat characteristics [20]. According to Markazi et al. [36], probiotics can decrease abdominal fat and improve bone development and growth in poultry. Lactic acid-producing bacteria may interfere with cholesterol absorption in the gut by producing bile salt hydrolase, which helps excrete more bile acids in the feces and reduce fat levels in meat [32].

Broilers fed a diet containing *L. lactis* JNU 534, which includes a high proportion of probiotics, demonstrated better slaughter performance than the control group. This improvement is likely due to the strain’s strong survivability in the gastrointestinal tract, which effectively prevents *Salmonella* colonization and stimulates the development of intestinal structures, enhancing the absorption and utilization of nutrients. Probiotics can improve intestinal performance through their antioxidant activity by inhibiting the production of oxygen free radicals, enhancing antioxidant capacity, and preventing oxidative damage to intestinal villi [37]. A study by Smialek et al. [38] showed that *L. lactis* can reduce the invasion of another pathogenic bacterium, *Campylobacter*, into the gastrointestinal tract of poultry under commercial production conditions, subsequently decreasing its presence in the gastrointestinal tract and carcasses after processing.

The exact mechanism by which feed probiotics impact meat quality remains unclear [39]. pH is a key qualitative factor of meat, playing a crucial role in influencing protein behavior in both fresh and processed meat products. It is also an important determinant of meat quality, affecting tenderness, color, and shelf life. In this study, the pH values for breast meat, which ranged from 5.75 to 5.82, fall within the accepted pH range for commercial poultry meat [40]. However, the results of this study do not align with some previous studies that reported probiotics can enhance meat quality. For instance, supplementation with the probiotic *Bacillus subtilis* has been associated with higher color lightness, greater water-holding capacity, a trend toward less cooking loss, and lower pH values [41]. These discrepancies may be due to the use of different bacterial strains as probiotics.

In animal bodies, probiotics can increase the activity of enzymes such as superoxide dismutase and glutathione peroxidase [22], which effectively reduce the levels of reactive oxygen species (ROS). This reduction in ROS helps maintain the freshness of myoglobin in fresh meat by minimizing the detrimental effects on muscle cell membranes containing phospholipids [42]. As a result, the color of the meat improves significantly, and drip loss is reduced [43]. The results obtained using different types and doses of probiotics administered in feed are variable and not always comparable.

## 5. Conclusions

These results indicate that *L. lactis* JNU 534 could act as a promising candidate for commercial broiler farms against the adverse effects on growth performance caused by salmonella without causing health issues. Moreover, it is important to underline that *L. lactis* JNU 534 supplementation showed improved nutrient utilization, growth efficiency, gut health, reduced abdominal fat deposition, and lowered cholesterol. Although the results show promising potential for *L. lactis* JNU 534 in the poultry industry, extending the research to other types of livestock could help confirm its wider use as an alternative to antibiotics. More comprehensive studies are needed to verify its effectiveness, safety, and potential in the feed and food industry.

## Figures and Tables

**Table 1 microorganisms-13-00525-t001:** Feed ingredients and nutrient composition of broiler basal diets during the starter, grower and finisher.

Item	Basal Starter	Basal Grower	Basal Finisher
Ingredients (%)			
Corn	52.2	55.9	61.7
Wheat bran	2.7	2.5	2.1
Soybean meal	34.1	31.6	27.2
Corn gluten meal	4.5	3.8	2.9
Soybean oil	2.5	2.5	2.5
Limestone	1.0	1.0	1.0
Dicalcium phosphate	1.3	1.25	1.1
Salt	0.25	0.25	0.25
L-Lysine	0.1	0.1	0.1
DL-methionine	0.25	0.2	0.15
Premix	1.0	1.0	1.0
Nutrient composition			
ME (kcal/kg)	3052	3104	3154
CP (%)	23.1	21.3	19.4
Lysine (%)	1.22	1.1	1.0

ME = Metabolizable energy; CP = Crude protein.

**Table 2 microorganisms-13-00525-t002:** The effect of *Lactococcus lactis* JNU 534 supplementation on weight gain, daily feed intake, and feed conversion ratio of broiler challenged with *Salmonella enteritidis*.

Item		Normal	Probiotic	SEM	*p*-Value
Body weight gain (BW, g/chick)					
	Day 0–7	172.82	182.01	1.61	0.37
	Day 8–14	402.62	417.45	4.27	0.24
	Day 15–21 *	753.81	799.89	4.75	0.00
	Day 21–28 *	1253.17	1316.39	5.66	0.01
	Day 28–35 *	1883.82	1941.70	9.12	0.03
Daily feed intake (gram/chick/day)					
	Day 0–7	17.78	17.18	0.325	0.801
	Day 8–14	49.89	50.72	0.427	0.392
	Day 15–21	89.15	88.75	1.830	0.862
	Day 21–28	122.48	120.53	3.261	0.595
	Day 28–35 *	163.38	182.27	4.526	0.007
Feed conversion ratio (g feed/g BW)					
	Day 0–7	0.69	0.67	0.017	0.896
	Day 8–14	1.16	1.13	0.012	0.116
	Day 15–21 *	1.44	1.37	0.019	0.025
	Day 21–28 *	1.56	1.48	0.026	0.034
	Day 28–35 *	1.64	1.34	0.041	0.000

* in a row, means are significantly different (*p* < 0.05), Independence sample *t*-test; SEM = Standard error of means.

**Table 3 microorganisms-13-00525-t003:** The effect of *Lactococcus lactis* JNU 534 supplementation on the blood characteristics of broiler challenged with *Salmonella enteritidis*.

Item	Blood Characteristics			
	Normal	Probiotic	SEM	*p*-Value
Creatinine (mg/dL)	0.22	0.23	0.014	0.863
Total bilirubin (mg/dL)	0.41	0.56	0.022	0.041
Cholesterol (mg/dL) *	112.87	99.00	1.407	0.000
Triglyceride (mg/dL) *	72.63	93.00	3.605	0.022
Free fatty acid (µEq/L)	461.50	389.25	45.16	0.246
Glucose (mg/dL) *	208.25	231.36	4.891	0.014
White blood cells (10^5^/µL)	21.51	23.16	4.591	0.810
Red blood cells (10^6^/µL)	6.37	6.28	0.177	0.728
Haemoglobin (g/dL)	13.14	12.99	0.278	0.687
Haematocrit (%)	52.30	52.91	1.709	0.776
Segment (%)	29.35	30.60	2.65	0.744
Lymphocyte (%)	59.00	59.34	3.68	0.950
Monocyte (%)	5.93	4.39	0.843	0.173
Eosinophil (%)	3.30	3.15	0.648	0.886
Basophil (%)	2.43	2.53	0.897	0.931

* in a row, means are significantly different (*p* < 0.05), Independence sample *t*-test; SEM = Standard error of means.

**Table 4 microorganisms-13-00525-t004:** The effect of *Lactococcus lactis* JNU 534 supplementation on internal organ weight of broiler challenged with *Salmonella enteritidis*.

Item	Internal Organ Weight (% Live Weight)			
	Normal	Probiotic	SEM	*p*-Value
Carcass *	79.98	81.51	0.38	0.020
Breast *	17.40	19.45	0.50	0.003
Leg	11.38	11.48	0.30	0.783
Thigh	6.43	6.47	0.29	0.912
Drumstick	4.95	5.02	0.19	0.785
Wing	3.97	3.85	0.11	0.474
Abdominal fat *	1.54	1.17	0.12	0.016
Liver	1.86	1.90	0.08	0.769
Heart	0.56	0.52	0.02	0.305
Gizard	1.37	1.34	0.07	0.797
Proventriculus	0.50	0.47	0.04	0.669
Duodenum	0.47	0.41	0.03	0.071
Jejenum	1.00	0.96	0.08	0.795
Ileum *	0.90	0.74	0.05	0.031
Secum	0.64	0.56	0.06	0.331
Colon	0.14	0.13	0.14	0.418
Pancreas	0.19	0.18	0.01	0.598
Limfa	0.08	0.10	0.02	0.425
Bursa	0.19	0.18	0.03	0.591

* in a row, means are significantly different (*p* < 0.05), Independence sample *t*-test; SEM = Standard error of means.

**Table 5 microorganisms-13-00525-t005:** The effect of *Lactococcus lactis* JNU 534 supplementation on meat characteristics of broiler challenged with *Salmonella enteritidis*.

Item	Meat Characteristics			
	Normal	Probiotic	SEM	*p*-Value
PH				
Breast	5.82	5.75	0.52	0.271
Leg	6.47	6,41	0.03	0.122
Water holding capacity (%)				
Breast	62.41	61.00	1.33	0.490
Leg	66.20	66.63	1.64	0.811
Cooking loss (%)				
Breast	10.90	9.84	0.61	0.301
Leg	10.84	11.22	0.71	0.653
Drip loss (%)				
Breast	6.17	5.63	0.84	0.612
Leg	2.01	2.17	0.11	0.464

SEM = Standard error of means.

## Data Availability

The original contributions presented in this study are included in the article. Further inquiries can be directed to the corresponding author.

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
