# Peer review of "Effect of Lactococcus lactis JNU 534 Supplementation on the Performance, Blood Parameters and Meat Characteristics of Salmonella enteritidis Inoculated Broilers"

_microorganisms, 2025, doi:10.3390/microorganisms13030525_

Round 1
Reviewer 1 Report
Comments and Suggestions for Authors
This manuscript explored the effects of the probiotic Lactococcus lactis JNU 534 on the growth performance,blood parameters,and meat characteristics of broilers infected with Salmonella.Salmonellosis in broilers is a disease that causes significant economic losses to the poultry industry and poses a risk to public health due to potential cross-contamination.The manuscript concluded that the supplementation of L.lactis JNU 534 showed improvements in nutrient utilization,growth efficiency,gut health,reduction of abdominal fat deposition,and lowering of cholesterol levels. This manuscript can provide some theoretical significance for practical production,but the manuscript has the following shortcomings in the following aspects.
1 A total of 96 one-day-old Arbor Acres broiler chickens, comprising both sexes were challenged with SE and randomly assigned into two treatment groups, housed in eight pens (four pens per each treatment, with 12 birds per pen). Each treatment only has 4 replicates,and under normal circumstances,the minimum number of replicates should be 6. Why not set up an antibiotic group as a positive control?
2 The two dietary treatment groups consisted of a control group receiving commercial feed, and a treatment group receiving commercial feed supplemented with 0.3% L. lactis JNU 534. What is the basis for the 0.3%addition of L.lactis JNU 534? Please provide a detailed explanation in the text. Please supplement the composition and nutritional level of the commercial feed.
3 A total of 96 Arbor Acres, one-day-old broiler chickens comprising both sexes were challenged with Salmonella enteritidis at concentration 1x109 CFU/ml on the third day. How can it be confirmed that all the chickens were infected with Salmonella?
4 The content in the introduction section of the manuscript is relatively scant.It is recommended to supplement with specific hazards of Salmonella to the poultry industry,specific measures taken in actual production,and the reasons or advantages for choosing L.lactis JNU 534 in this study.
5 After removing the skin, feathers, viscera, head, and feet, the carcass weight was measured. Please supplement its implementation standards. The same question with the water holing capacity and cooking loss.
6 Please add data statistical methods in the Materials and Methods section.
7 The results section should simply and clearly state the results without adding any other unrelated content.It is suggested to remove L103-106.
8 L117 The study showed that L. lactis JNU 534 did not significantly affect (p>0.05) the…. Please check here and everywhere in the result section.
9 The discussion section does not elaborate on how L.lactis JNU 534 specifically affects the growth performance and meat quality of Salmonella-infected broilers;the mechanism of impact should be analyzed in conjunction with the results of this experiment.
10 L171 remove P value.
11 please add DOI for each reference.
Author Response
Title: Effect of Lactococcus lactis JNU 534 supplementation on the Performance, Blood Parameters and Meat Characteristics of Salmonella enteritidis inoculated broilers
- Comments: A total of 96 one-day-old Arbor Acres broiler chickens, comprising both sexes were challenged with SE and randomly assigned into two treatment groups, housed in eight pens (four pens per each treatment, with 12 birds per pen). Each treatment only has 4 replicates,and under normal circumstances, the minimum number of replicates should be 6. Why not set up an antibiotic group as a positive control?
Response 1: Thanks for the suggestion. I agree with the comment, however we did not set up antibiotic group due to ban on the use of antibiotics as growth promoters in animal feed.
- Comments: The two dietary treatment groups consisted of a control group receiving commercial feed, and a treatment group receiving commercial feed supplemented with 0.3% lactis JNU 534. What is the basis for the 0.3% addition of L.lactis JNU 534? Please provide a detailed explanation in the text. Please supplement the composition and nutritional level of the commercial feed.
Response 1: Thanks for the suggestion. The basis for the 0.3% addition of L.lactis JNU 534 due to the antibiotic growth promotor banned nowadays is in the range of 2.5 – 3% on feed. We tested this of L.lactis JNU 534 on substitution the use of antibiotic to prevent the salmonella infection. L. lactis also have the antibacterial compounds called bacteriocin. Bacteriocin is a natural proteinaceous substance that inhibits the growth of pathogenic bacteria. Several barriers may be involved and affect their stability and biological activity. Since bacteriocins are polypeptides (small-sized proteins), it is assumed that proteolytic enzymes in the stomach and other parts of the intestinal tract can hydrolyze them, rendering them inert. We supplement the composition and nutritional level of the commercial feed in table 1. The revised manuscript this change can be found in page 2-3 line 80-85 and page 6 (179-186).
- Comments: A total of 96 Arbor Acres, one-day-old broiler chickens comprising both sexes were challenged with Salmonella enteritidis at concentration 1x109 CFU/ml on the third day. How can it be confirmed that all the chickens were infected with Salmonella?
Response 1: Thanks for the comment. We confirmed that all chickens were infected by checking the visual excreta of the chicken, other symptoms that we can see in the chicken is weakness, loss of appetite and poor growth. In this study, the growth performance was lower than commercial broilers (line 75-77)
- Comments: The content in the introduction section of the manuscript is relatively scant.It is recommended to supplement with specific hazards of Salmonella to the poultry industry,specific measures taken in actual production,and the reasons or advantages for choosing L.lactis JNU 534 in this study.
Response 1: Thanks for the suggestion. We revise the introduction by supplement with specific hazards of Salmonella to the poultry industry,specific measures taken in actual production, and the reasons or advantages for choosing L.lactis JNU 534 in this study. The main reason for use this probiotic due to a new bacteriocin-producing lactic acid bacteria isolated from kimchi was identified as Lactococcus lactis JNU 534, presenting preservative properties for foods of animal origin. The revised manuscript this change can be found in page 1 line 40-43, page 2 line 52-54; 56-64.
- Comments: After removing the skin, feathers, viscera, head, and feet, the carcass weight was measured. Please supplement its implementation standards. The same question with the water holing capacity and cooking loss.
Response 1: Thanks for the suggestion. This method was measured according to Wandita et al, 2018.
- Comments: Please add data statistical methods in the Materials and Methods section.
Response 1: Thanks for the suggestion. In this revise file, we include the data statistical methods in the Materials and Methods section. The revised manuscript this change can be found in page 3 line 118-121.
- Comments: The results section should simply and clearly state the results without adding any other unrelated content.It is suggested to remove L103-106.
Response 1: Thanks for the suggestion. We remove L103-106 (In recent decades, the role of probiotics in enhancing poultry growth performance and health has been well documented. Probiotics are closely linked to the host's immune system. A critical challenge lies in selecting potential probiotic strains that can successfully function in the gastrointestinal tract (GIT) and withstand various stressors.)
- Comments: L117 The study showed that L. lactis JNU 534 did not significantly affect (p>0.05) the…. Please check here and everywhere in the result section.
Response 1: Thanks for the suggestion. We check everywhere in the result section and revise this manuscript carefully.
- Comments: The discussion section does not elaborate on how L.lactis JNU 534 specifically affects the growth performance and meat quality of Salmonella-infected broilers;the mechanism of impact should be analyzed in conjunction with the results of this experiment.
Response 1: Thanks for the suggestion. We add more discussion to prove the result. The revised manuscript this change can be found in page 6 line 162-165; 179-186 and page 7 line 207-209; 217-220.
- Comments: L171 remove P value
Response 1: Thanks for the suggestion. We removed the P value in that line.
- Comments: please add DOI for each reference
Response 1: Thanks for the suggestion. We revise all the reference by adding the DOI

Reviewer 2 Report
Comments and Suggestions for Authors
The manuscript written by Listya Purnamasari, Joseph F. dela Cruz, Dae Yeon Cho, Kwang Ho Lee, Sung Min Cho, Seung Sik Chung, Yong Jun Choi, Jun Koo Yi and Seong Gu Hwang is a good manuscript that describes the effect of Lactococcus lactis supplements on broilers inoculated with Salmonella enteritidis.
The text is clear, using comprehensive English. I encourage the publication of the manuscript after minor changes outlined below:
L 16 - I suggest a rephrase „Probiotics have been proposed as alternative feed additives to antibiotics”. Even though antibiotics are currently used in broilers, I think the authors should not put the antibiotics in the food additive section, instead elaborate the dangerous effects of antibiotic usage in animal and human health, in contrast with the positive effects of probiotics.
L 102 – Were there any microbiological tests performed to assess the changes in the gut microbiota after the usage of L. lactic? Like the effect on S. enteritidis?
L 113 – In Table 1 please describe abbreviated term in legend (SEM), check for similar errors.
L 145 – The Discussion section could be a bit improved.
Author Response
Title: Effect of Lactococcus lactis JNU 534 supplementation on the Performance, Blood Parameters and Meat Characteristics of Salmonella enteritidis inoculated broilers
- Comments: L 16 - I suggest a rephrase „Probiotics have been proposed as alternative feed additives to antibiotics”. Even though antibiotics are currently used in broilers, I think the authors should not put the antibiotics in the food additive section, instead elaborate the dangerous effects of antibiotic usage in animal and human health, in contrast with the positive effects of probiotics.
Response 1: Thanks for the suggestion. We change line 16 by deleting the word “antibiotics”.
- Comments: L 102 – Were there any microbiological tests performed to assess the changes in the gut microbiota after the usage of L. lactic? Like the effect on S. enteritidis?
Response 1: Thanks for the comment. For confirmed that all chickens were infected by checking the visual excreta of the chicken, other symptoms that we can see in the chicken is weakness, loss of appetite and poor growth. In this study, the growth performance was lower than commercial broilers. The revised manuscript this change can be found in page 2 line 75-77. However, we didn’t perform microbial test to assess the changes in the gut microbiota after the usage of L. lactic. In the future research we will considering this suggestion.
- Comments: L 113 – In Table 1 please describe abbreviated term in legend (SEM), check for similar errors.
Response 1: Thanks for the suggestion. We describe abbreviated term in legend (SEM) in all of the tables.
- Comments: L 145 – The Discussion section could be a bit improved.
Response 1: Thanks for the comment. We add more discussion to prove the result. The revised manuscript this change can be found in page 6 line 162-165; 179-186 and page 7 line 207-209; 217-220.

Round 2
Reviewer 1 Report
Comments and Suggestions for Authors
The number of replicates in the experimental design is too small. This study only has 4 replicates per treatment group, which is relatively low, and the minimum generally required is 6.
Author Response
Title: Effect of Lactococcus lactis JNU 534 supplementation on the Performance, Blood Parameters and Meat Characteristics of Salmonella enteritidis inoculated broilers
- Comments: L72-74: How were the broilers inoculated with SE? Gavage, in feed, in water?
Response 1: Thanks for the suggestion. broiler chickens were challenged with Salmonella enteritidis at concentration 1x109 CFU/ml on the third day orally. (L72-74)
- Comments: L73-76: Please comment on why 4 replicates were used instead of 6 (as per comment by reviewer #1)
Response 1: Thanks for the suggestion. The basis for the 0.3% addition of L.lactis JNU 534 due to the antibiotic growth promotor banned nowadays is in the range of 2.5 – 3% on feed. Due to constraints at the poultry experimental farm, we reduced the number of replicates to four instead of six. To ensure reliable results, we increased the number of animals per replicate.
- Comments: L78-81: Supplementation with “0.3% L. lactis JNU 534” does not quantify the level of L. lactis used…. What does 0.3% L. lactis translate into CFU/gm? The authors quantified the level of SE that was inoculated (1 x 109 cfu/ml) and should do the same for the probiotic strain.
Response 1: Thanks for the comment. The 0.3% Lactococcus lactis in the study translates to approximately 1-2 x 109 CFU per gram. While we did not directly quantify the bacterial load, the manufacturer of the product guarantees that the L. lactis used falls within this range.
- Comments: L98: Change ‘centrifuge’ to ‘centrifuged’; state centrifugation parameters as relative centrifugal force (RCF) stated as xg , not as RPM.
Response 1: Thanks for the suggestion. We revise the word and the centrifuge unit
- Comments: L100: ‘Parameters’ is listed twice; eliminate one of them.
Response 1: Thanks for the suggestion. It deleted
- Comments: L113-117: Again, the RCF (xg) should be listed instead of RPM; also, mention that the tube was decanted before weighing (to determine cooking loss).
Response 1: Thanks for the suggestion. We revise the centrifuge unit
- Comments: L182-185: Should be reworded: The basis for the 0.3% addition of L. lactis JNU 534 as a replacement for antibiotic growth promotants (now banned) that were used in the 2.5-3% range in feed. (the 0.3% should be quantified in cfu/gm parameters).
Response 1: Thanks for the suggestion. The 0.3% Lactococcus lactis in the study translates to approximately 1-2 x 109 CFU per gram. While we did not directly quantify the bacterial load, the manufacturer of the product guarantees that the L. lactis used falls within this range.
Comments: L185-190: Does L. lactis JNU 534 produce a bacteriocin? Not all strains of L. lactis produce bacteriocins. Also, the authors bring up the production of bacteriocins as potential inhibitor of SE, here and earlier; however most bacteriocins produced by lactic acid bacteria do not inhibit Gram (-) bacteria and the authors provide no evidence that it does. The authors should temper such suggestions unless supported by data or citations in the literature that bacteriocins of L. lactis inhibit Salmonella Enteritidis (make sure it’s the bacteriocin and not the lactic acid that they also produce). If there could be any implication for inhibition, lactic acid is a known inhibitor of SE and the authors could implicate that aspect.
Response 1: Thanks for the suggestion we supported some more citations in the literature that bacteriocins of L. lactis inhibit Salmonella Enteritidis
